# The Cerato-Platanin EPL2 from *Trichoderma reesei* Is Not Directly Involved in Cellulase Formation but in Cell Wall Remodeling

**DOI:** 10.3390/microorganisms11081965

**Published:** 2023-07-31

**Authors:** Alinne Costa Silva, Letícia Harumi Oshiquiri, Luiz Felipe de Morais Costa de Jesus, David Batista Maués, Roberto do Nascimento Silva

**Affiliations:** Department of Biochemistry and Immunology, Ribeirão Preto Medical School, University of São Paulo, Ribeirão Preto 14049-900, SP, Brazil; alinnecs09@hotmail.com (A.C.S.); leticia_oshiquiri@hotmail.com (L.H.O.); lfmoraess21@gmail.com (L.F.d.M.C.d.J.); dbmaues@gmail.com (D.B.M.)

**Keywords:** CPs, *Trichoderma*, fungi, cell wall synthesis, stressors, cellulolytic genes, hydrolases, transcriptome

## Abstract

*Trichoderma reesei* is a saprophytic fungus that produces large amounts of cellulases and is widely used for biotechnological applications. Cerato-platanins (CPs) are a family of proteins universally distributed among Dikarya fungi and have been implicated in various functions related to fungal physiology and interaction with the environment. In *T. reesei*, three CPs are encoded in the genome: *Trire2_111449*, *Trire2_123955,* and *Trire2_82662*. However, their function is not fully elucidated. In this study, we deleted the *Trire2_123955* gene (named here as *epl2*) in the wild-type QM6aΔ*tmus53*Δ*pyr4* (WT) strain and examined the behavior of the Δ*epl2* strain compared with WT grown for 72 h in 1% cellulose using RNA sequencing. Of the 9143 genes in the *T. reesei* genome, 760 were differentially expressed, including 260 only in WT, 214 only in Δ*epl2*, and 286 in both. Genes involved in oxidative stress, oxidoreductase activity, antioxidant activity, and transport were upregulated in the Δ*epl2* mutant. Genes encoding cell wall synthesis were upregulated in the mutant strain during the late growth stage. The Δ*epl2* mutant accumulated chitin and glucan at higher levels than the parental strain and was more resistant to cell wall stressors. These results suggest a compensatory effect in cell wall remodeling due to the absence of EPL2 in *T. reesei*. This study is expected to contribute to a better understanding of the role of the EPL2 protein in *T. reesei* and improve its application in biotechnological fields.

## 1. Introduction

*Trichoderma reesei* is a saprophytic fungus present in different habitats [1]. Due to its capacity to produce cellulolytic enzymes, it has excellent biotechnological applications [2]. In the presence of inducing carbon sources, such as cellulose, *T. reesei* produces high levels of cellulases that degrade the substrate. The sugars generated are transported across the cytoplasmatic membrane and metabolized [3]. Despite being a widely studied microorganism, how *T. reesei* detects and recognizes the external environment, such as the available carbon sources, metal ions, and pH, is not fully elucidated [3].

Proteins of the cerato-platanin (CP) family are small fungal-exclusive proteins found in both Ascomycota and Basidiomycota [4]. They are related to the early development of fungi and fungus–host interactions [5]. These proteins are found abundantly in secretomes during hyphal growth, mycelial mass formation, sporulation, and spore maturation [6]. They are involved in cell wall remodeling and enlargement during hyphal growth and chlamydospore formation [7]. CPs are characterized by four cysteines forming two disulfide bridges and can be identified by the cerato-platanin domain (IPR010829). Structural analyses have demonstrated the similarity of CPs with the D1 domain of plant expansins, which are related to the recognition and binding of polysaccharides [8]. CPs are located in the fungal cell wall and are secreted. Accordingly, 95% of the identified CPs are predicted to have a signal peptide. Additionally, it was shown that CPs have approximately 120 amino acids, with an average of 37% hydrophobic residues, giving them hydrophobic properties [4].

In phytopathogenic fungi, CPs act as effectors that suppress basal plant defense. This role in virulence has been observed in *Botrytis cinérea* [9]*, Sclerotinia sclerotiorum* [10], and *Magnaporthe grisea* [11]. However, CPs of mycoparasitic fungi, such as *Trichoderma atroviride* [12]*, Trichoderma virens* [13], and *Trichoderma harzianum* [14] have shown the opposite effect. They trigger plant immune defense and are therefore also known as eliciting plant response-like (EPL) proteins. The proposed mechanism suggests that *Trichoderma* spp. can regulate the hydrophobicity of the host or substrate surfaces, making them more hydrophilic to facilitate access to nutrients or plant interactions [15].

In the *Trichoderma reesei* genome (mycocosm.jgi.doe.gov/Trire2/Trire2.home.html, accessed on 20 June 2021), three genes encode CPs: *Trire2_123955*, *Trire2_111449*, and *Trire2_82662*. In previous stu/dies, we identified that the gene *Trire2_123955* is upregulated in cellulose, and the gene *Trire2_82662* is downregulated in the presence of glucose [16]. Thus, considering the biotechnological importance of this fungus and the study of the potential physiological role of the CP proteins, in this work, we aimed to understand the function of CP proteins in the production of cellulases and the development of the fungus *T. reesei*. To accomplish this, we used an RNA sequencing (RNA-Seq) approach to perform a large-scale comparative analysis of the transcriptomes of the *T. reesei* strains QM6a∆*tmus53*∆*pyr4* (wild-type) and Δ*epl2* (mutant strain lacking the gene *Trire2_123955*) grown in the presence of cellulose. Our findings provide valuable insights into the physiological role of CP proteins in *T. reesei* and can potentially contribute to the development of more efficient methods for the application of this fungus in biotechnology fields.

## 2. Materials and Methods

### 2.1. Strains and Culture Conditions

The *T. reesei* strain QM6aΔ*tmus53*Δ*pyr4* [17] was obtained from the Research Area Gene Technology and Applied Biochemistry, Institute for Chemical Engineering, Vienna University of Technology, Austria, and used here as the wild-type (WT) strain. The strain was maintained at 4 °C on malt extract agar (MEA) medium (malt extract (Neogen, Neogen, MI, USA) 3% (*w*/*v*) and bacteriological agar (Kasvi, Madrid, Spain) 2% (*w*/*v*)) supplemented with 5 mM uridine (Sigma-Aldrich, St. Louis, MO, USA) due to the lack of the gene *pyr4* in the WT. To conduct the experiments, the WT strain and the Δ*epl2* strain (obtained in this study) were grown on MEA medium for 7 to 10 days until complete sporulation.

For the gene expression assays, 10^6^ conidia/mL from the WT and Δ*epl2* strains were inoculated into Mandels–Andreotti (MA) medium (2 g/L KH_2_PO_4_ (Synth, Diadema, SP, Brazil), 1.4 g/L (NH_4_)_2_SO_4_ (Synth, Diadema, SP, Brazil), 0.3 g/L MgSO_4_·7H_2_O (Synth, Diadema, SP, Brazil), 0.4 g/L CaCl_2_ (Sigma-Aldrich, St. Louis, MO, USA), 0.1 M Na_2_HPO_4_·2H_2_O (Synth, Diadema, SP, Brazil), 0.1 M citric acid (Dinâmica, São Paulo, SP, Brazil), 5 mg/L FeSO_4_·7H_2_O (Synth, Diadema, SP, Brazil), 1.7 mg/L MnSO_4_·H_2_O (Synth, Diadema, SP, Brazil), 1.4 mg/L ZnSO_4_·7H_2_O (Synth, Diadema, SP, Brazil), 2 mg/L CoCl_2_·2H_2_O (Synth, Diadema, SP, Brazil), 0.3 g/L urea (Synth, Diadema, SP, Brazil), and 1 g/L peptone (Kasvi, Madrid, Spain), pH 5.0) supplemented with glycerol (Synth, Diadema, SP, Brazil) 1% (*v*/*v*) for 24 h at 30 °C and 200 rpm. After this period, the mycelia were collected and washed with MA medium without carbon sources. Then, the same amount of mycelium was transferred to a fresh MA medium supplemented with the cellulose Avicel (Synth, Diadema, SP, Brazil) 1% (*w*/*v*) and incubated at 30 °C and 200 rpm for the indicated time.

### 2.2. Cladogram Construction

The amino acid sequences of proteins characterized or cited as CPs were extracted from the NCBI database [18]. Similarly, amino acid sequences of *Trichoderma* spp. proteins predicted as CPs were obtained from the Joint Genome Institute (JGI) [19]. The sequences were aligned using MAFFT [20], and the cladogram was inferred using FastTree 2 [21]. The complete list of proteins, including their identifications, organisms, and references, is available in Appendix A.

### 2.3. T. reesei epl2 Gene Deletion

To delete the *epl2* gene, the *epl2* open reading frame (ORF) was replaced with the selection marker *pyr4* encoding the orotidine-5′-phosphate decarboxylase enzyme. The complete deletion cassette was assembled using *S. cerevisiae* SC9721 (genotype: *MATa his3*Δ*200 ura3–52 leu2*Δ*1 lys2*Δ*202 trp1*Δ*63*) by homologous recombination [22]. To transform *S. cerevisiae*, the 5′ UTR and 3′ UTR regions of the *epl2* gene and the *pyr4* gene were amplified from *T. reesei* DNA (Appendix A) and, together with the linearized pRS426 vector, were used to transform the yeast. The resulting deletion cassette was amplified by PCR and used to transform *T. reesei* QM6aΔ*tmus53*Δ*pyr4* (WT).

### 2.4. Transformation of T. reesei

The transformation of the *T. reesei* WT was performed as described by Gruber and coworkers [23]. A total of 20 μg of linear DNA from the deletion cassette was used in the assay. After transformation, the plates were incubated at 30 °C for 3 to 4 days until spores were visible. The candidates were submitted to three rounds of selection on MA medium with and without 0.1% Triton X-100 (GE Healthcare, Uppsala, Sweden). Cassette integration in the genome locus was verified by PCR using a specific set of primers for *pyr4* and *epl2* (Appendix A). The expression profile of the *epl2* gene was also analyzed by RT-qPCR (Appendix A).

### 2.5. RNA Extraction and Transcript Analysis Using Quantitative PCR (RT-qPCR)

The mycelia from the WT *T. reesei* strain and the Δ*epl2* strain grown on cellulose for 24, 48, and 72 h (after being pre-grown in glycerol for 24 h) were filtered through Miracloth (Merck, Darmstadt, Hesse, Germany), frozen in liquid nitrogen, and macerated. According to the manufacturer’s instructions, the total RNA was isolated using TRI Reagent (Sigma-Aldrich, St. Louis, MO, USA). The samples were quantified, and their integrity was verified using a 1% agarose gel.

To analyze gene expression by RT-qPCR, 1 μg of RNA was treated with DNAse I (ThermoFisher Scientific, Waltham, MA, USA) to remove genomic DNA. After DNA removal, cDNA synthesis was performed using a Maxima First Strand cDNA Synthesis Kit (ThermoFisher Scientific, Waltham, MA, USA) according to the manufacturer’s instructions. cDNA was diluted in water (1:50) and used for RT-qPCR analysis in Bio-Rad CFX96TM equipment using SsoFast EvaGreen Supermix (Bio-Rad, Hercules, CA, USA) for the detection signal according to the manufacturer’s instructions. The actin gene was used for endogenous control to normalize the total amount of cDNA in each reaction. The genes analyzed are represented in Appendix A.

### 2.6. RNA Sequencing

To perform the sequencing, RNA from the WT and Δ*epl2* strains grown in glycerol 1% for 24 h and cellulose 1% for 72 h (MA medium) was isolated in biological triplicate. For growth in cellulose, the strains were pre-cultured in 1% glycerol for 24 h and then transferred to a new flask containing fresh medium with 1% cellulose. The RNAs obtained from the WT and Δ*epl2* strains of *T. reesei* were diluted in water and quantified, and 20 μg was treated with the Qiagen RNAeasy Kit (QIAGEN, Hilden, North Rhine-Westphalia, Germany) according to the manufacturer’s recommendations. Sequencing was performed by Novogene Corporation Inc. in Sacramento, California, USA, using Illumina technology.

### 2.7. Data Analysis

FastQC v0.11.5 [24] was used to assess the quality of the data obtained by sequencing. The removal of adapters was performed with Cutadapt v3.4 [25] using the following configuration: removal of reads with N > 10%, Qscore ≤ 5, and minimum size < 150 bp. Sequences were mapped based on the *Trichoderma reesei* v2.0 reference genome provided by the Joint Genome Institute (JGI) Genome Portal (http://genome.jgipsf.org/Trire2/Trire2.home.html, accessed on 20 June 2021) using STAR v2.7.9 [26]. HTSeq-count v0.13.5 [27] was used to count the mapped reads, and expression values were calculated using the DESeq package version 1.31.21 [28]. The genes present in this analysis were annotated using InterProScan version 5.51-85 [29]. Differential expression was represented in log2 fold change, and genes were considered upregulated when the log2 fold change was ≥1 and downregulated when the log2 fold change was ≤−1 using an adjusted *p*-value < 0.05 as a threshold). In addition, to identify the genes differentially expressed in Δ*epl2* in relation to WT in cellulose, we subtracted the log_2_ fold change values. A difference of at least 1 was set as the cut-off. A Venn diagram was created using the Euler package in R [30]. In accordance with Wright and coworkers, functional categorization was performed with Gene Ontology (GO) terms [31].

### 2.8. Cell Wall Polysaccharide Quantification

About 10 mg of dry cell wall mass was used for polysaccharide extraction, in accordance with François [32]. After extraction, 1 mL from the final preparation was concentrated 10× by lyophilization. The sugars were analyzed using HPLC (Young Lin YL9100 series system) by a refractive index (RI) detector at 40 °C. The samples were loaded in the REZEX ROA (Phenomenex, Torrance, CA, USA) column (300 × 7.8 mm) at 85 °C and eluted with 0.05 M sulfuric acid at a flow rate of 1.5 mL/min [33]. The saccharides were quantified based on a standard curve (5–100 mM) using HPLC standards from Sigma-Aldrich (St. Louis, MO, USA): Tetraacetylchitotetraose (Chit4), triacetylchitotriose (Chit3), diacetyl-β-D-chitobiose (Chit2), N-Acetil-β-D-glucosamina (nag), and glucose (gluc).

### 2.9. Test of Sensitivity to Cell Wall Stressors

To assess how the fungal strains respond to cell wall stressors, 10 µL of a solution containing 10^6^ conidia/mL from the WT and Δ*epl2* strains of *T. reesei* were inoculated on the center of minimal media (MM) plates (1 g/L MgSO_4_·7H_2_O (Synth, Diadema, SP, Brazil), 10 g/L KH_2_PO_4_ (Synth, Brazil), 6 g/L (NH_4_)_2_SO_4_ (Synth, Diadema, SP, Brazil), 3 g/L Na_3_C_6_H_5_O_7_ (Synth, Diadema, SP, Brazil), 20 mL/L trace elements, and 20 g/L bacteriological agar (Kasvi, Madrid, Spain)) with 2% glucose (Synth, Diadema, SP, Brazil) in the absence or presence of different concentrations of Calcofluor White (CFW) (Sigma-Aldrich, St. Louis, MO, USA) (20 and 40 µg/mL) or Congo Red (CR) (Sigma-Aldrich, St. Louis, MO, USA) (100 and 200 µg/mL). After incubation for 3 days, the mycelia diameter under the indicated conditions was recorded. Three replicates of each experiment were conducted. Statistical tests were performed using one-way analysis of variance (ANOVA) followed by Bonferroni’s test (available in Prism software v8.0) for comparing the growth of the parental and mutant strains. CFW and CR stains were used for chitin and β-1,3-glucan; the former interacts preferentially with chitin, and both interfere with cell wall assembly and integrity [34]. The use of these stressors allows the study of fungi strains with a role in cell wall maintenance.

### 2.10. Microscopy Analysis

The methodology used for the cell wall thickness analysis was in accordance with Ribeiro et al. [33], with some modifications. The WT and the mutant strain Δ*epl2* were cultivated in microculture dishes using the modified Riddell technique [35]. Briefly, conidia from WT and Δ*epl2* were inoculated into small blocks (~1 cm^2^) of MEA medium between sterile coverslips for 48 h at 30 °C. After growing the cultures, the coverslips were removed from the culture dish, stained with 0.001% Calcofluor White (Sigma-Aldrich, St. Louis, MO, USA) for 5 min, and washed three times with phosphate-buffered saline (PBS) 1×. Coverslips were inspected on a Leica SP5 confocal microscope using a 63× objective oil immersion lens. For the Calcofluor White (CFW) stain, the excitation wavelength was 405 nm, and the emission wavelength was 433 nm. The differential interference contrast (DIC) and confocal images were captured with PMTs and HyD detectors and processed using the software LAS AF 2.7.3.9723. After the images were acquired, they were submitted for wall thickness analysis using the ImageJ software (https://imagej.nih.gov/ij/index.html, accessed on 31 March 2023). Statistical tests were performed using one-way analysis of variance (ANOVA) followed by Bonferroni’s test (available in Prism software v8.0) to compare the cell wall thickness of the parental (*n* = 42) and mutant (*n* = 39) strains.

## 3. Results

### 3.1. Classification of Cerato-Platanins from Trichoderma reesei

We used the amino acid sequences of 25 CPs to classify the cerato-platanins (CPs) in *T. reesei* according to the studies [7,9,10,11,12,36,37,38,39,40,41,42] (Appendix A). Figure 1 shows that *Trichoderma* spp. proteins are divided into four clades: EPL1 (SM1), EPL2 (SM2), EPL3 (SM3), and EPL4 (SM4). Some of these proteins have been analyzed or characterized, such as EPL1 from *Trichoderma atroviride* [12,42], Epl1-Tas from *Trichoderma asperellum* [39], Epl-1 from *Trichoderma harzianum* [14,40], and SM1 from *Trichoderma virens* [12,13,37], in the EPL1 clade; EPL-2 from *T. atroviride* [42], SM2 from *T. virens* [38], and the protein analyzed in this study (EPL2 from *Trichoderma reesei*), in the EPL2 clade; EPL3 from *T. atroviride* [42] and SM3 from *T. virens* [38], in the EPL3 clade; and SM4 from *T. virens* in the EPL4 clade. Proteins from plant pathogens such as *Fusarium graminearum* and *Sclerotinia sclerotiorum* were grouped into other clades (Figure 1).

### 3.2. Deletion of the epl2 Gene in T. reesei

To obtain a better understanding of the function of the cerato-platanin protein from *T. reesei*, the ORF of the *epl2* gene was replaced with the auxotrophic marker gene *pyr4* in the strain QM6aΔ*tmus53*Δ*pyr4*, referred to here as the WT strain, resulting in the Δ*epl2* strain.

Confirmation of the absence of the *epl2* gene (753 pb) (Figure 2A) and the *pyr4* gene insertion (3.246 pb) was performed by PCR using specific primers (Appendix A). The correct integration of the deletion cassette in the Δ*epl2* strain was also verified by PCR using primers that bind at 200 bp before the 5′UTR region and 200 bp after the 3′UTR region. The results showed the presence of amplicons with the expected size (Figure 2B). Strains 8, 9, 11, 12, and 14 show bands corresponding to the expected fragment sizes (Figure 2B). In addition, no *epl2* mRNA was detected by RT-qPCR (Figure 2C). These results confirmed the complete deletion of the *epl2* gene from the fungal genome.

### 3.3. Global Analysis of the Genes Regulated by Δepl2 in the Presence of Cellulose

The WT and Δ*epl2* strains of *T. reesei* were initially grown in biological triplicate in a medium containing cellulose 1% (*w*/*v*) as the sole carbon source. The global transcriptome analysis was performed using RNA-Seq. The preparation of the cDNA libraries generated many reads. They were processed and mapped to the *Trichoderma reesei* v2.0 genome (https://mycocosm.jgi.doe.gov/Trire2/Trire2.home.html, accessed on 31 March 2023). Overall, 98% of the reads were mapped to the reference genome (Appendix A), and a principal component analysis demonstrated the reliability of the RNA-Seq samples (Appendix A).

The global expression of the Δ*epl2* and WT strains revealed that of the 9143 *T. reesei* genes, 2338 were differentially expressed in the WT, and 2074 were expressed in Δ*epl2* (cellulose in relation to glycerol) (Appendix A). After the screening, 760 genes were considered differentially expressed between these strains, with 260 unique to WT (166 upregulated and 97 downregulated), 214 unique to Δ*epl2* (177 upregulated and 37 downregulated), and 286 in common for both strains (252 upregulated and 33 downregulated) (Figure 3).

### 3.4. Top Genes Regulated in the Δepl2mutant in the Cellulose Condition

The upregulated and downregulated genes in the WT and Δ*epl2* were categorized using GO terms. To verify the biological processes (Appendix A), molecular functions (Appendix A), and cellular components (Appendix A) involved in cellulose culture, a hierarchical tree cluster analysis was performed. The most enriched biological processes for both strains in cellulose in relation to glycerol are the glucosamine-containing metabolic processes, amino sugar metabolic processes, oxidative stress responses, and lipid-catabolic processes. Among the upregulated processes for Δ*epl2* are organic acid transport, oxidative stress responses, oxidoreductase activity, antioxidant activity, and transportation.

The ten genes most upregulated and downregulated in the Δ*epl2* strain compared to WT after cellulose culture are listed in Table 1. The gene with the highest expression encodes a protein of the glucose–methanol–choline oxidoreductase family, known to have a FAD-binding domain and a substrate-binding domain, which can be various sugars, alcohols, cholesterol, and choline [43]. The most downregulated gene encodes a long-chain fatty acid Acyl-CoA ligase protein, which is involved in lipid metabolism. This enzyme forms Acyl-CoA, allowing fatty acids to move into the β-oxidation pathway [44]. It should be noted that several genes that encode proteins with unknown functions were found to be both up- and downregulated. Furthermore, a gene that encodes for a candidate protein for glucan degradation was upregulated in the Δ*epl2* mutant, indicating the participation of EPL2 in cell wall remodeling (Table 1).

### 3.5. Expression of Carbohydrate-Active Enzyme (CAZy) Genes

Cellulases from *T. reesei* are classified as endoglucanases, exoglucanases, or cellobiohydrolases (CBH), and β-glucosidases. Hakkinen and coworkers organized a new annotation of all genes in the CAZy family and identified 201 genes for glycosyl hydrolases (GH), 22 for carbohydrate esterases (CE), and 5 for polysaccharide lyases [45]. This databank is continuously updated at the site http://www.cazy.org/, accessed on 13 November 2021. We constructed a table containing all CAZy family genes upregulated in Δ*epl2* (Table 2). Of the nine upregulated genes, eight belong to the GH family, and one belongs to the CE family. The gene encoding a candidate β-1,3-1,4-glucanase (ID 123726) showed the most significant fold change. Other genes that showed positive regulation were the polysaccharide monooxygenase enzyme Cel61 a (ID 73643), xylanase 2 (ID 123818), N-acetyl glucosamine (GH89—ID 69700), some candidate glucanases (ID 54242, 123639), N-acetylglucosaminidase (ID 69700), and mannosidase (ID 60635).

We also decided to analyze the expression of cellulase genes using RT-qPCR at different times of cultivation in the presence of cellulose. As can be seen in Figure 4, the genes that encode for β-glucosidase (*cel3a*) (Figure 4A), xyloglucanase (*cel74a*) (Figure 4B), and exoglucanase (*cel6a*) (Figure 4D) are downregulated in the mutant Δ*epl2* in relation to the WT. On the other hand, the *cel7a* gene that encodes for exoglucanase is upregulated in the mutant Δ*epl2* in relation to the WT (Figure 4C). However, the up- or downregulation of cellulase genes in the Δ*epl2* mutant is not reflected in the cellulolytic enzyme activity compared to the WT (Appendix A). The results suggest that EPL2 does not act directly in regulating cellulase formation. It seems that the *epl2* gene is fundamental for the cell wall remodeling process [46,47].

### 3.6. EPL2 Is Involved in Cell Wall Remodeling

To investigate the influence of EPL2 in cell wall remodeling, we first evaluated the growth and sensitivity of the WT and Δ*epl2* strains in the presence of cell wall stressors such as Calcofluor White (CFW) and Congo Red (CR) (Figure 5)**.** As shown in Figure 5, the Δ*epl2* strain was more tolerant to CFW and CR stressors even at concentrations as high as 40 µg/mL and 200 µg/mL, respectively (Figure 5B).

Tolerance to the cell wall damage caused by CFW and CR indicates that the Δ*epl2* strain can alter the fungal cell wall’s composition and organization. Therefore, we investigated whether the deletion of *epl2* affects cell wall organization in *T. reesei.* Confocal microscopy analysis using CFW showed that *epl2* deletion causes an increase in cell wall thickness, with cell wall thicknesses of 0.35 ± 0.05 μm for the mutant and 0.27 ± 0.025 μm for the WT strain (*p <* 0.01) (Figure 6A).

Furthermore, we performed RT-qPCR using genes involved in fungal cell wall remodeling (Appendix A) and sugar quantification by HPLC. For gene expression, we selected genes involved in cell wall remodeling (N-acetyl-glucosaminidases, chitinases, and glucanases) differentially expressed in the RNA-Seq data and assessed their profile in WT and Δ*epl2* strains grown in cellulose. Our results revealed a significant increase in the expression of all genes in the Δ*epl2* strain at the final cultivation time (72 h) compared to the WT strain (Figure 6B). In contrast, the WT strain showed the highest gene expression levels at 48 h of cultivation. This result suggests a compensation response to the absence of EPL2 protein. Notably, the *Trire2_59791* and *Trire2_54242* genes, which encode chitinase and glucanase proteins, displayed higher expression levels during the early hours of cultivation in the WT strain. In contrast, the Δ*epl2* strain showed higher expression levels of these two genes at the end of cultivation.

Furthermore, the gene-encoding glucan endo-1,3-α-glucosidase (*Trire2_71532*) showed increased expression levels during the cultivation of the Δ*epl2* strain, unlike the WT strain. In contrast, both *T. reesei* strains exhibited low expression levels of the *Trire2_108672* and *Trire2_73643* genes encoding glucanase and β-1,6-N-acetylglucosaminyltransferase, respectively (Figure 6B). The fungal cell wall comprises polysaccharides, including chitin, β-1,3-glucan, and β-1,6-glucan. Polysaccharides from the cell wall were extracted and hydrolyzed, and HPLC analyzed the composition of fungal cell walls from the WT and Δ*epl2* strains. The HPLC results showed that the deletion of *epl2* alters the composition of the cell wall, causing an increase in chitin oligomers (chit4, chit3, and chit2), N-acetyl-glucosamine (Nag), and glucose (Figure 6C). These results showed that EPL2 has an essential role in cell wall integrity maintenance in *T. reesei*, affecting the expression of genes involved in cell wall remodeling and participating in cell wall composition and organization.

## 4. Discussion

CPs are found exclusively in fungi and are involved in growth, development, recognition, adhesion, cell wall morphogenesis, and parasitism [42,48,49,50]. Despite several studies of CPs in fungi, these proteins’ biological function remains unclear in *T. reesei*.

In this work, we constructed a cladogram that showed four clades for *Trichoderma* spp. In the EPL1 (SM1) clade, all the proteins were shown to be involved in plant resistance. We found that EPL1 from *Trichoderma atroviride*, which forms protein layers, increases the polarity of aqueous solutions and surfaces, binds to chitin, and induces plant growth and resistance to phytopathogens [12,42]; Epl1-Tas from *Trichoderma asperellum* elicits woody plant resistance to pathogens [39]; Epl-1 from *Trichoderma harzianum* is involved in mycoparasitism, self-recognition, and the expression of defense-related genes in plants [14,39]; and SM1 from *Trichoderma virens* induces the production of reactive oxygen species and elicits the expression of defense-related genes in plants [13,37].

In the EPL2 clade, we also identified proteins involved in plant resistance. EPL-2 from *T. atroviride* is slightly expressed during cultivation with glucose and strongly expressed during cultivation on colloidal chitin or fungal conidiation. However, its knock-out strain shows no effect on diverse growth and development tests, including biomass formation on shake flasks, agar plate growth, hydrophobicity, osmotic stress, and cell wall stress [42]. There is also SM2 from *T. virens,* which is not expressed on shake flask cultivation with glucose but is expressed during conidiation on potato dextrose agar plates. Similar to EPL2 from *T. atroviride*, its knock-out strain shows no phenotypic difference from the parental strain; however, both proteins have been shown to be involved in plant protection [38]. In addition, this clade includes the protein analyzed in this study: EPL2 from *T. reesei*.

In the EPL3 clade, we identified EPL3 from *T. atroviride*, which was described by Frischmann et al. [42]. However, its expression was not detected in most of the analyzed situations, including during cultivation with colloidal chitin or stages of spore formation. In this clade, we also observed SM3 from *T. virens*, the transcript of which was not detected during shake flask cultivation with glucose, nor was it detected on potato dextrose agar plates [38].

In the EPL4 clade, we identified proteins exclusively from *T. atroviride*, *T. virens*, and *T. asperellum*, consistent with the study conducted by Gao and colleagues [15]. Furthermore, we observed that the proteins from plant pathogens were grouped in different clades. This observation aligns with previous findings that these proteins play a role in virulence [6,9,10,11], a characteristic not present in *Trichoderma* spp. (Figure 1).

To better understand the function of *epl2* in *T. reesei*, we deleted this gene in the QM6a∆*tmus53*∆*pyr4* strain, referred to here as the WT. To gain novel insights into the role of this gene in *T. reesei* physiology, we performed a global expression analysis comparing the *T. reesei* Δ*epl2* mutant strain with its WT strain. The functional categorization of genes differentially expressed in the Δ*epl2* strain demonstrated processes related to transport, oxidoreductase activity, polysaccharide binding, and carbohydrate binding. Considering that *T. reesei* was subjected to a hydrolytic enzyme-inducing condition (i.e., growth in cellulose), these processes can be related to recognizing the carbon source present in the medium, suggesting that the absence of the EPL2 protein could cause an increase in activities associated with the extracellular medium. Nevertheless, no role has so far been reported for the EPL2 protein in *T. reesei*.

Studies demonstrated that CPs could bind to and weaken lignocellulosic materials, acting similarly to plant expansins and facilitating access to cellulolytic enzymes [50,51]. This characteristic has been reported in the CP of *Ceratocystis populicola* [51], *Moniliophthora perniciosa* [52], *Fusarium graminearum* [53], and *Trichoderma harzianum* [54]. In *T. reesei*, CPs were found in the secretome during fermentation processes in sugarcane bagasse [55]. To assess the influence of *epl2* deletion on cellulose recognition and degradation, a CAZymes analysis was performed. Nine genes were shown to be upregulated in the Δ*epl2* mutant, indicating that EPL2 may be involved in cellulase gene expression. In addition, we conducted a temporal analysis of the expression of the main cellulase genes in both the WT and Δ*epl2* strains when grown on 1% cellulose at 24, 48, and 72 h. Although we observed differences in gene expression between the two strains, our enzymatic activity assays and protein dosage showed no significant difference between them. This suggests that the role of EPL2 may primarily occur at the transcriptional level.

Interestingly, our results indicated that EPL2 plays a role in cell wall remodeling, as the Δ*epl2* strain showed tolerance to cell wall stressors, increased cell wall thickness, upregulation of genes involved in cell wall remodeling, and the composition of polysaccharides from the cell wall. It seems that the absence of the *epl2* gene generates a compensatory effect in the fungus, which starts to express some enzymes responsible for synthesizing cell wall components in greater quantity. Thus, we observed higher chitin, glucan, and Nag content. This is in accordance with *T. reesei*’s remarkable capacity of adapting to changing environmental conditions [56]. One way that *T. reesei* achieves this is by remodeling its cell wall, which involves the deposition, degradation, and modification of different cell wall components, including chitin, glucans, and proteins [57].

Although CPs are abundant in fungal secretomes, they have not been studied systematically in the genus *Trichoderma*. In some *Trichoderma* species, the genes for CPs are present at higher levels during the early processes of fungal development. In *T. atroviride*, *epl2* expression occurred during hyphae growth and mycelium development [42]. Similarly, the *sm1* gene of *T. virens* was expressed in germinating spores. Nevertheless, studies performed with strains deleted for *epl2* did not show differences in growth, conidia germination, and hyphae development [13,37]. In *T. reesei*, deleting the *epl2* gene did not cause significant development changes in a cellulose medium.

In summary, our results suggest the involvement of EPL2 in cell wall remodeling and the growth of *T. reesei* and demonstrate that it is indirectly related to increasing cellulase production. More studies are necessary to explain the real role of CPs in cellulase production/recognition. The three CPs in *T. reesei* probably work together in different stages of development and hydrolase production.

## 5. Conclusions

Our analysis of the *T. reesei* transcriptome identified several genes that were affected by the *epl2* deletion, including cellulolytic genes and genes related to cell wall remodeling. However, most genes were indirectly influenced by *epl2*. Our results also suggest that the EPL2 protein could be involved in cellulose recognition processes and cell wall remodeling. Further study of the two other cerato-platanins from *T. reesei* may provide additional insights into the function of these proteins in this fungus and their potential applications in biotechnology.

## Figures and Tables

**Figure 1 microorganisms-11-01965-f001:**
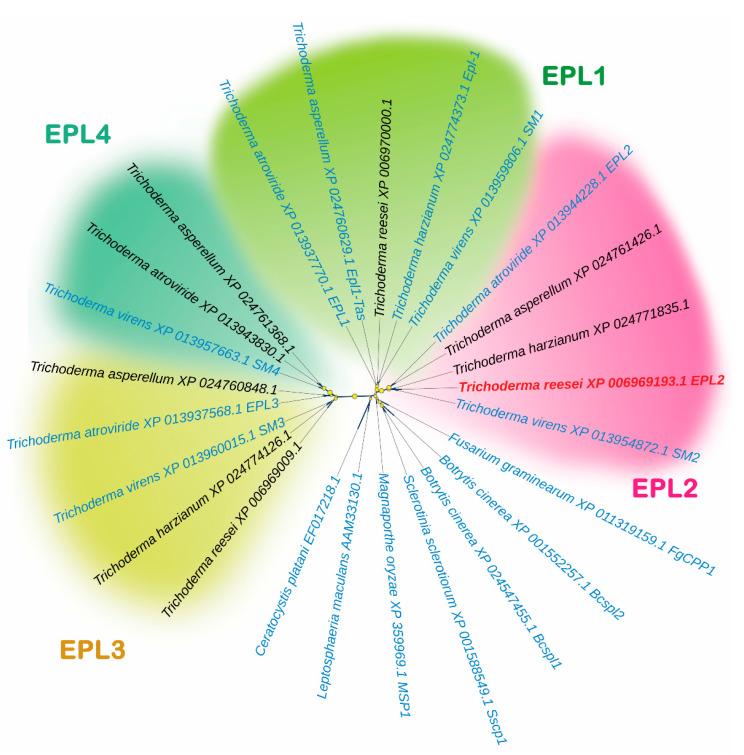
Approximate maximum likelihood cladogram of cerato-platanins present in *Trichoderma* spp. and phytopathogenic fungi. Three clades were formed for *Trichoderma* spp.: EPL1, EPL2, EPL3, and EPL4. Cerato-platanins of pathogenic fungi are represented outside of the colored clades. Identifiers in blue represent proteins characterized or described in the literature, while the protein under study is highlighted in red. Yellow circles indicate support values greater than 0.8, obtained after running 1000 replicates. The amino acid sequences used in this analysis can be obtained according to their NCBI ID.

**Figure 2 microorganisms-11-01965-f002:**
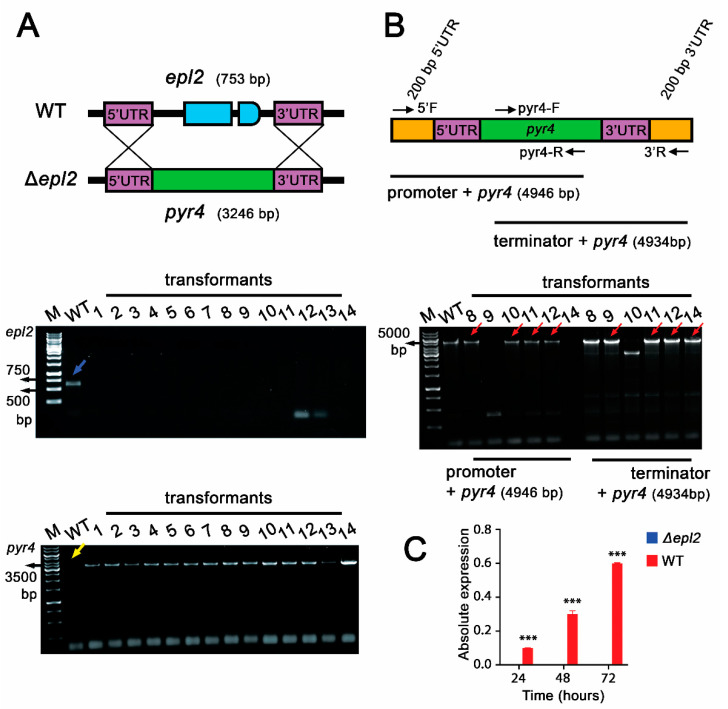
Confirmation of obtaining *T. reesei* Δ*epl2* mutant: (**A**) Deletion strategy and PCR amplification of the coding region *epl2*, validating the creation of the Δ*epl2* mutant strain and PCR amplification of the molecular marker (*pyr4*) during the validation of the Δ*epl2* mutant strain. The blue arrow indicates the coding region of *epl2* only in the WT strain. The yellow arrow indicates the absence of the gene *pyr4* in the WT strain. (**B**) PCR of Δ*epl2* mutant strains 5′UTR + *pyr4* region and *pyr4* + 3′UTR region. The red arrows indicate the fragments including 5′UTR + pyr4 and pyr4 + 3′UTR in the mutant strains. (**C**) RT-qPCR analysis of *epl2* used to validate *epl2* deletion. WT: Wild-type positive control. M: Molecular weight marker (O’GeneRuler 1 Kb DNA Ladder, Thermo Fisher Scientific, Waltham, MA, USA). *** = *p* ≤ 0.001.

**Figure 3 microorganisms-11-01965-f003:**
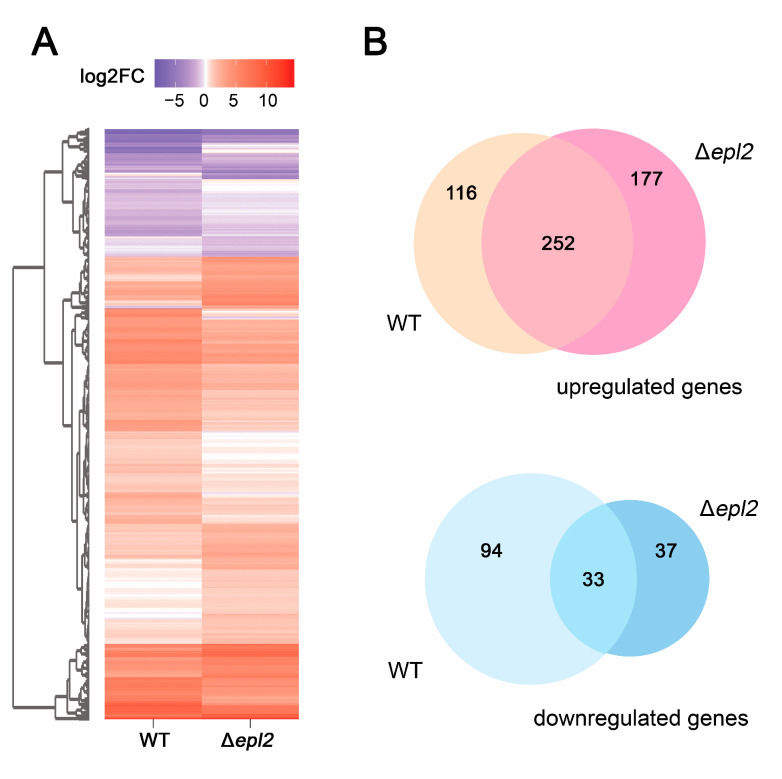
Gene expression in wild-type (WT) and mutant Δ*epl2* in cellulose in relation to glycerol, where blue tones indicate downregulation and red tones indicate upregulation: (**A**) Heat map accounting for 760 genes. (**B**) Venn diagram indicating the common elements between the strains in the study.

**Figure 4 microorganisms-11-01965-f004:**
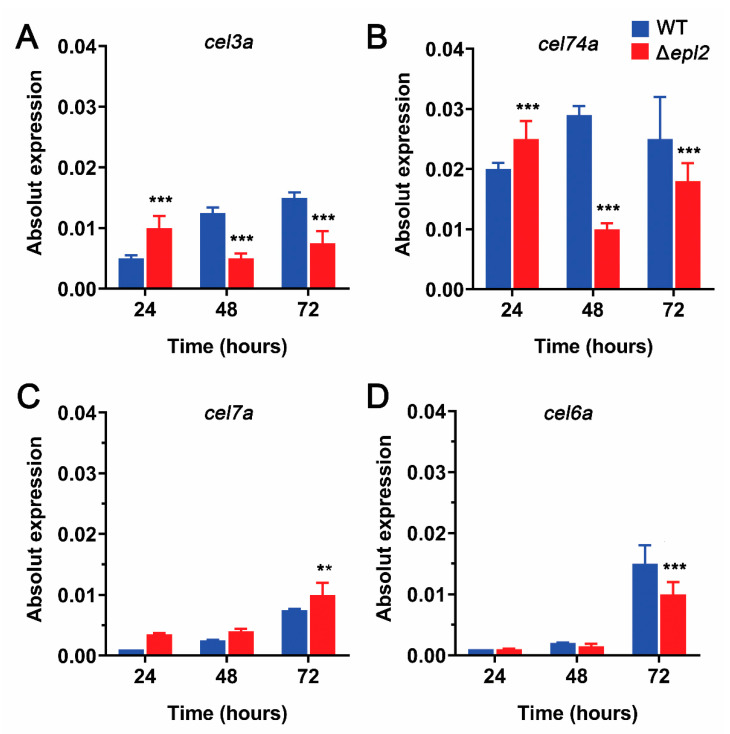
Cellulase gene expression in WT and Δ*epl2* mutant grown on cellulose for 24, 48, and 72 h: (**A**) β-glucosidase (*cel3a*). (**B**) Xyloglucanase (*cel74a*). (**C**) Exoglucanase (*cel7a*). (**D**) Exoglucanase (*cel6a*). The values presented are the means of three groups of experiments, with the corresponding deviations. For statistical analysis, a two-way ANOVA test was performed (95% confidence interval). Asterisks indicate groups that showed significant differences when compared with the WT strain. ** = *p* ≤ 0.01. and *** = *p* ≤ 0.001.

**Figure 5 microorganisms-11-01965-f005:**
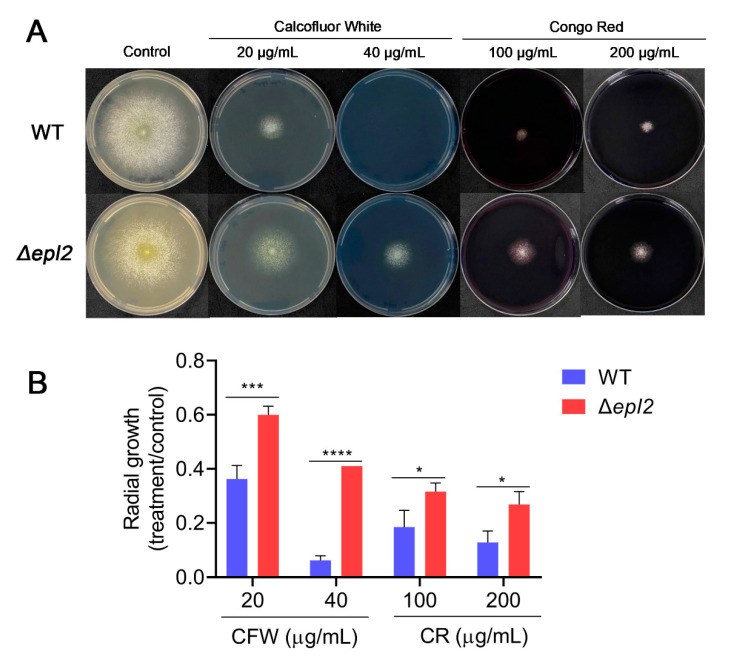
Cell wall stressor analysis. *T. reesei* WT and Δ*epl2* strains were inoculated on the center of MM plates with 2% glucose in the absence or presence of different concentrations of Calcofluor White (CFW) (20 and 40 µg/mL) or Congo Red (CR) (100 and 200 µg/mL): (**A**) Images registered after 3 days of growth. (**B**) Radial growth. The measures were taken in centimeters (cm), and the results are expressed as radial growth of treatment/radial growth of control (without any stressor). For statistical analysis, a one-way analysis of variance (ANOVA) followed by Bonferroni’s test was performed (95% confidence interval). Asterisks indicate groups that showed significant differences when compared with the WT. * = *p* ≤ 0.05, *** = *p* ≤ 0.01, and **** = *p* ≤ 0.001.

**Figure 6 microorganisms-11-01965-f006:**
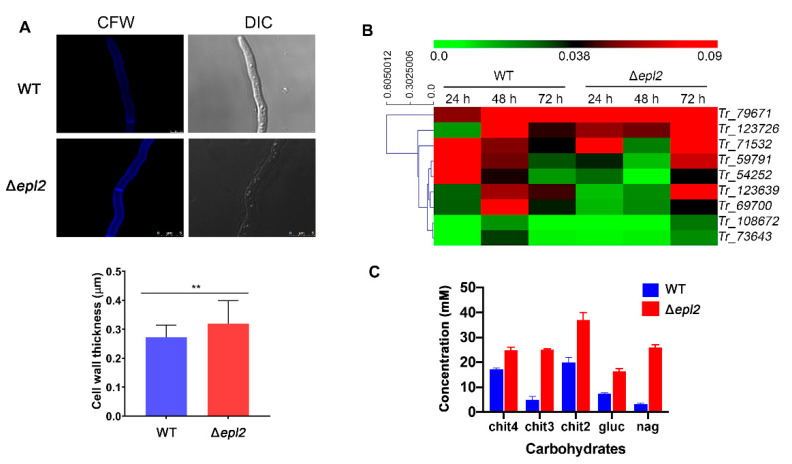
Deletion of *epl2* affects cell wall organization in *T. reesei*: (**A**) Cells were stained with Calcofluor White, visualized using confocal microscopy (top panel), and measured to compare the cell wall thickness (bottom panel). (**B**) Gene expression analysis of genes involved in fungal cell wall remodeling by RT-qPCR. WT and Δ*epl2* strains were grown in MA medium with cellulose 1% (*w*/*v*) for the indicated time, and then the RNA was extracted. (**C**) Concentration of sugars in the cell wall. For statistical analysis, a one-way analysis of variance (ANOVA) followed by Bonferroni’s test was performed (95% confidence interval). Asterisks indicate groups that showed significant differences when compared with the WT. *Trire2_* is abbreviated as *Tr_* in the figure. Tetraacetylchitotetraose (Chit4), triacetylchitotriose (Chit3), diacetyl-β-D-chitobiose (Chit2), N-Acetil-β-D-glucosamina (nag), and glucose (gluc). ** = *p* ≤ 0.01.

**Table 1 microorganisms-11-01965-t001:** Description of the ten most upregulated and downregulated genes in Δ*epl2* grown in cellulose in relation to the WT. Gene expression values are presented as log_2_ fold change (FC).

Protein ID	InterPro Description	FC	*p*-Value
80659	Glucose–methanol–choline oxidoreductase	14.56	3.92 × 10^−30^
105808	NA	11.08	3.98 × 10^−19^
123726	Candidate β-1,3-1,4-glucanase	10.54	7.78 × 10^−54^
111890	Phosphate transporter	10.26	5.94 × 10^−24^
70972	MFS transporter	10.14	1.01 × 10^−15^
111716	NAD-dependent epimerase/dehydratase	9.95	6.00 × 10^−15^
121785	NA	9.02	7.23 × 10^−91^
112147	NA	8.94	2.00 × 10^−46^
58472	Catalase	8.90	9.44 × 10^−46^
109146	NA	8.55	6.25 × 10^−11^
66999	AMP-binding enzyme	−7.23	3.55 × 10^−7^
65067	NADP-dependent oxidoreductase	−5.74	0.0010
105242	S-adenosyl-L-methionine-dependent methyltransferase	−5.51	0.0031
76682	ABC-transporter	−5.21	0.0005
105220	Mycotoxin biosynthesis protein UstYa-like	−4.48	0.0092
105488	NA	−4.25	0.0017
70500	NA	−4.24	0.0042
70838	NA	−4.19	0.0004
3914	Glycosyl transferase CAP10 domain	−4.12	0.0013
32027	A/β hydrolase fold	−3.60	0.0156

NA = not available.

**Table 2 microorganisms-11-01965-t002:** CAZy genes upregulated in Δ*epl2* mutant, relative to the expression in WT, grown in cellulose. Gene expression values are presented as log_2_ fold change (FC).

Protein ID	Class	Family	Annotation	FC	*p*-Value
123726	Glycoside hydrolase	16	Candidate β-1,3-1,4-glucanase	10.54	7.78 × 10^−54^
123639	Glycoside hydrolase	64	Candidate β-1,3-glucanase	8.16	1.10 × 10^−8^
123818	Glycoside hydrolase	11	Xylanase 2	5.98	0.001
73643	Glycoside hydrolase	61	Endoglucanase	5.81	0.0142
69700	Glycoside hydrolase	89	α-N-acetylglucosaminidase	3.97	0.0024
71532	Glycoside hydrolase	71	Glucan endo-1,3-α-glucosidase	3.67	4.42 × 10^−108^
60635	Glycoside hydrolase	92	Candidate α-1,2-mannosidase	2.98	0.0020
79671	Carbohydrate Esterase	9	Candidate N-acetyl-glucosamine-6-phosphate deacetylase	2.89	8.14 × 10^−42^
54242	Glycoside hydrolase	55	Candidate β-1,3-glucanase	1.80	1.71 × 10^−6^

## Data Availability

All the relevant data are included in this manuscript. Additional data are available on request from the corresponding author.

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
