# Peer review of "The Cerato-Platanin EPL2 from *Trichoderma reesei* Is Not Directly Involved in Cellulase Formation but in Cell Wall Remodeling"

_microorganisms, 2023, doi:10.3390/microorganisms11081965_

Round 1
Reviewer 1 Report
Thework is logical and focussed
The paper requires extensive editing to allow better reading and understanding -see many sticky notes
more background information in Introduction would have helped
Also diagram of a typical Trich cell wall would aid clarity

starts off well but becomes problematical with depth into paper
several sections are quite muddled
Author Response
- Lines 56-60.
Dear reviewer, thanks for the observation. The name has already corrected and the adjustments have been made.
- Line 71. “what was the source of cellulose used in study?”
Dear reviewer, thanks for the comment. We added the information about the source of cellulase that we used in the cultivations. “For gene expression assays, 106 conidia/mL from WT and Δepl2 strains were inoculated into Mandels-Andreotti (MA) medium (2 g/L KH2PO4, 1.4 g/L (NH4)2SO4, 0.3 g/L MgSO4·7H2O, 0.4 g/L CaCl2, 0.1 M Na2HPO4·2H2O, 0.1 M citric acid, 5 mg/L FeSO4·7H2O, 1,7 mg/L MnSO4·H2O, 1.4 mg/L ZnSO4·7H2O, 2 mg/L CoCl2.2H2O, 0.3 g/L urea and 1 g/L peptone, pH 5.0) supplemented with glycerol (Synth) 1% (v/v) for 24 h at 30 °C and 200 rpm. After this period, the mycelia were collected, washed with MA medium without carbon sources and then the same amount of mycelia was transferred to fresh MA medium supplemented with cellulose Avicel (Synth) 1% (w/v) and incubated at 30 °C and 200 rpm for the indicated time. “
- Line 71. “what does this encode?”
Dear reviewer, thanks for the observation. We rewrote the sentence and added this information to the manuscript. “To delete the epl2 gene, a deletion cassette was constructed by replacing the gene of interest with the epl2 open reading frame (ORF) was replaced with the selection marker pyr4, encoding the the orotidine-5′-phosphate decarboxylase enzyme”.
- Line 86. “is QM6a the wt strain then please clarify”
Dear reviewer, thanks for the comment. Yes, the QM6aΔtmus53Δpyr4 strain, sometimes referred as simply QM6a, was used as the WT strain. To avoid any doubt, now we used the term Wild-type (WT) along the full text, instead of QM6a, QM6aΔtmus53Δpyr4 or parental strain.
- Line 71. “what age of culture? what culture phase?”
Dear reviewer, thanks for the observation. We rewrote the sentence and included this information. “The mycelia from T. reesei WTstrain QM6a and Δepl2 strains grown on cellulose for 24, 48 and 72 h (after pregrown in glycerol for 24 h) were filtered through Miracloth, frozen in liquid nitrogen, and macerated.”
- Line 134. “please provide sentences describing what these dyes are collecting”
Dear reviewer, thanks for the suggestion. We added the information to the manuscript. “CFW and CR stain for chitin and β-1,3-glucan, while the first interacts preferentially with chitin and both interfere with cell wall assembly and integrity, allowing the study of fungi strains with role in cell wall maintenance”
Line 142. “what was media space before h”
Dear reviewer, thanks for the observation. We added the information about the media used for cultivation for microscopy analysis and corrected the time unit. “The WT parental strain QM6a and the mutant strain Δepl2 were cultivated for 48h in microculture dishes using the modified Riddell technique [34][33]. Briefly, conidia from WT and Δepl2 were inoculated into small blocks (~1 cm2) of MEA medium between sterile coverslips for 48 h at 30 °C.”
Line 157. “based on studies for genes in what microbes Please add information”
Dear reviewer, thanks for the observation. The protein ID and its respective organism used in the phylogenetic analysis are listed in Table S1, now referred in the text, and we specify them throughout the text.
“TTo construct the cladogram and classify the Cerato-platanins (CPs) present in T. reesei, we used the sequence of amino acids of 23 fungal proteins, according to the studies [7–9,36–44] (Table S1). According to Figure 1, Trichoderma spp. proteins are divided into 3 clades: EPL1 (SM1), EPL2 (SM2) and EPL3 (SM3). Among these proteins, some have been characterized in the literature, such as EPL1 from Trichoderma atroviride [38,44], Epl1-Tas from Trichoderma asperellum [41], Epl-1 from Trichoderma harzianum [42] and SM1 from Trichoderma virens [38,39] in EPL1 clade, EPL-2 from T. atroviride [44], SM2 from T. virens [40] and the protein analyzed in this study (EPL2 from Trichoderma reesei) in EPL2 clade and EPL3 from T. atroviride [43] and SM3 from T. virens [40] in the EPL3 clade. Proteins from plant pathogens such as Fusarium graminearum and Sclerotinia sclerotiorum were grouped into other clades (Figure 1). “
Line 156-1853. “reword you are talking about a constructed mutant now in a specific strain describe”; “do not understand”; “new sentence, unclear as written”; “again too complex in writing needs reconstruction”; “whole section requires rewriting for clarity
this is too abbreviated to read well “
Dear reviewer, thanks for the comments. First, we moved this part for section Discussion. Here, we brought information about the CPs characterized in other fungi. Therefore, throughout the text we used the nomenclature of the protein established for the respective organism in which it was characterized. The sentences are constructed with the protein name, the organism of origin and the biological role of this protein in the respective fungus.
Line 166. “poor wording, redraft, the gene product not the gene is producing results?”
Dear reviewer, thanks for the comment. In the case for EPL1 from T. atroviride, the protein was purified from the fungal supernatant and biochemically characterized (https://doi.org/10.1074/jbc.M112.427633), getting the information mentioned in the text. In another study (https://doi.org/10.3389/fpls.2015.00077), this gene was overexpressed and deleted in this fungus, and novel functions were revealed. We rewrote the sentence in order to clarify this informations. “we found EPL1 from Trichoderma atroviride, which forms protein layers, increases the polarity of aqueous solutions and surfaces, binds to chitin, and induces plant growth and resistance to phytopathogens, and functional characterization of this gene by deletion and overexpression showed that EPL1 from T. atroviride is important for protection against phytophatogens and plant resistance”
Line 169. “of what?”
Dear reviewer, we do not understand your comment. If it's about fungal conidiation, this is the asexual reproduction of the fungi, whereas expression of epl-2 from T. atroviride is induced.
Line 172. “what is used as C sources here?”
Dear reviewer, thanks for the comment. The carbon source used in the experiments was glucose or glycerol 1% (https://doi.org/10.1074/jbc.M112.427633, https://doi.org/10.1016/j.fgb.2004.09.002)
Line 211. “what %”
Dear reviewer, thanks for observation. We included this information in the text. “T. reesei strains QM6aWT and Δepl2 were initially grown in biological triplicate in a medium containing cellulose 1% (w/v) as the sole carbon source.”
Line 227. “in common to both strains”
Dear reviewer, thanks for the observation. We changed that for your suggestion.
Line 268. “It is growth on cellulose not just presence
Dear reviewer, thanks for the observation. We changed that for your suggestion.
Line 280. “growth on cellulose? or anything else? needs clarification”
Dear reviewer, thanks for the observation. Yes, we referred to the growth in cellulose. We rewrote the sentence to make it clearer. “and it is known that the growth of T. reesei is closely related to the production of cellulases, where growth in cellulose is characterized by an early phase of high biomass production and low cellulase levels and a later, log phase of high cellulase production and low biomass production”
Line 290. “see earlier comments what do these materials show”
Dear reviewer, thanks for the comment. The adjustments have been made.
Line 301. “ explain why you think this”
Dear reviewer, we rewrote all this section to clarify the experiments we conducted to understand the role of epl2 in maintaining fungal cell wall integrity
Line 334. “I would have valued a clear description of the polymers in the cell walls of parental cells in the introduction DIAGRAMS would be valuable additions”. Figure 6. “need to explain all terms on x axis”
Dear reviewer, thanks for the suggestion. We included these information in the Results section regarding fungal cell wall composition. “Fungal cell wall is composed polysaccharides including chitin, β-1,3-glucan and β-1,6-glucan. Polysaccharides from cell wall were exctrated and hydrolyzed and the composition of fungal cell wall from WT and Δepl2 strains was analyzed by HPLC and the results showed that deletion of epl2 alters the composition of cell wall, causing an increase in chitin oligomers (chit4, chit3, and chit2), N-acetyl-glucosamine (Nag), and glucose”
Line 332. “but they have crossovers to plant enzymes yes?”
Dear reviewer, thanks for the comment. Indeed, some CPs may show expansin activity (https://doi.org/10.1016/j.plaphy.2019.03.025)
Line 339. “poor wording you mean growth on cellulose”
Dear reviewer, thanks for the comment. We emphasize this in the text now. “Considering that T. reesei was subjected to a hydrolytic enzyme-inducing condition – growth in cellulose –,”
Line 353. “are they lectin then?”
Dear reviewer, thanks for the comment. We are not able to found any study classifying CP as lectins.
Line 360. “which protein?”
Dear reviewer, thanks for the comment. We referred to the EPL2 protein. We added this information to the manuscript.
Line 386. “italics for gene”
Dear reviewer, thanks for the observation. We corrected the name in the manuscript.
Reviewer 2 Report
The authors of the manuscript microorganisms-2418105 investigate the function of cerato-platanin in Trichoderma reesei strain QM6a. The manuscript needs improvements before publication.
The Introduction section must include more details related to cerato-platanins (CP), their universal presence in Dikarya fungi (Chen H, Kovalchuk A, Keriö S, Asiegbu FO. 2013. Distribution and bioinformatic analysis of the cerato-platanin protein family in Dikarya. Mycologia 105,1479–1488) and their putative functions. The function of cerato-platanin in fungal cell wall remodeling was demonstrated from the very beginning of CP description – Baccelli, I., Comparini, C., Bettini, P. P., Martellini, F., Ruocco, M., Pazzagli, L., Bernardi, R. & Scala, A. (2012). The expression of the cerato-platanin gene is related to hyphal growth and chlamydospores formation in Ceratocystis platani. FEMS Microbiology Letters, 327(2), 155-163. The function mentioned in the paper of Gao, R., Ding, M., Jiang, S., Zhao, Z., Chenthamara, K., Shen, Q., F. Cai & Druzhinina, I. S. (2020). The evolutionary and functional paradox of cerato-platanins in fungi. Applied and Environmental Microbiology, 86(13), e00696-20 is not presented.
The EPL4 (nom. nov.; GenBank accession number XP_013943830), mentioned by the above paper of Gao et al., is not considered in 2.2. Phylogenetic tree construction. The EPL4 gene is present in a few Trichoderma strains; however, the authors must mention it and explain the reason for not considering it.
More details must be presented in sub-section 2.8. Cell wall polysaccharides quantification. Parrou and François's initial method is related to quantifying both polysaccharides (glycogen) and oligo/disaccharides (trehalose). Authors must briefly present the adaptation of this method to the fungal cell wall and how this method was used for quantification of chitin oligomers (chit4, chit3, and chit2), N-acetyl-glucosamine, and glucose.
Author Response
Comments and Suggestions for Authors
The authors of the manuscript microorganisms-2418105 investigate the function of cerato-platanin in Trichoderma reesei strain QM6a. The manuscript needs improvements before publication.
The Introduction section must include more details related to cerato-platanins (CP), their universal presence in Dikarya fungi (Chen H, Kovalchuk A, Keriö S, Asiegbu FO. 2013. Distribution and bioinformatic analysis of the cerato-platanin protein family in Dikarya. Mycologia 105,1479–1488) and their putative functions.
The function of cerato-platanin in fungal cell wall remodeling was demonstrated from the very beginning of CP description – Baccelli, I., Comparini, C., Bettini, P. P., Martellini, F., Ruocco, M., Pazzagli, L., Bernardi, R. & Scala, A. (2012). The expression of the cerato-platanin gene is related to hyphal growth and chlamydospores formation in Ceratocystis platani. FEMS Microbiology Letters, 327(2), 155-163.
The function mentioned in the paper of Gao, R., Ding, M., Jiang, S., Zhao, Z., Chenthamara, K., Shen, Q., F. Cai & Druzhinina, I. S. (2020). The evolutionary and functional paradox of cerato-platanins in fungi. Applied and Environmental Microbiology, 86(13), e00696-20 is not presented.
Response: Thank you for your comment. We have revised the introduction section to include more details about Cerato-platanins, their presence in Dikarya and their functions, with base on the suggested papers. The following paragraphs were modified:
“Proteins of the Cerato-platanin (CPs) family are small fungal-exclusive proteins found in both Ascomycota and Basidiomycota. They related to the early development of fungi as well as to fungus-host interactions [4]. These proteins are found abundantly in secretomes during hyphae growth, mycelial mass formation, sporulation, and spore mat-uration [5]. They have been shown to be involved in cell wall remodeling and enlargement during hyphal growth and chlamydospores formation [6]. CPs are characterized by the presence of four cysteines that form two disulfide bridges and can be identified by the cerato-platanin domain (IPR010829). Structural analyses demonstrated the similarity of CPs with the D1 domain of plant expansins, which are related to the recognition and binding of polysaccharides [7]. CPs are located in the fungal cell wall and are secreted, with 95% of identified proteins predicted to have a signal peptide. Additionally, it was shown that CPs have approximately 120 amino acids, with an average of 37% hydropho-bic residues, giving them hydrophobic properties [8],
In phytopathogenic fungi, CPs act as effectors that suppress basal plant defense. This role in virulence has been observed in Botrytis cinérea [9], Sclerotinia sclerotiorum [10] and Magnaporthe grisea [11]. However, CPs trigger plant immune defense in mycoparasitic fun-gi and are known as Eliciting plant response-like (EPL) proteins. This characteristic has been observed in fungal species, such as Trichoderma atroviride [12], Trichoderma virens [13] and Trichoderma harzianum [14]. The proposed mechanism is that Trichoderma spp. control the hydrophobicity of host or substrate surfaces, making them more hydrophilic to facilitate access to nutrients or plant interactions [15].”
The EPL4 (nom. nov.; GenBank accession number XP_013943830), mentioned by the above paper of Gao et al., is not considered in 2.2. Phylogenetic tree construction. The EPL4 gene is present in a few Trichoderma strains; however, the authors must mention it and explain the reason for not considering it.
Response: Thank you for your comment. In the first version of our manuscript, we didn’t include the epl4 because our focus was on T. reesei, and this gene is not present in its genome. However, after considering your feedback, we realized that including EPL4 would improve our analysis. We have constructed another cladogram that includes EPL4 and updated the manuscript to reflect this change. We appreciate your valuable contribution to improving our work.
More details must be presented in sub-section 2.8. Cell wall polysaccharides quantification. Parrou and François's initial method is related to quantifying both polysaccharides (glycogen) and oligo/disaccharides (trehalose). Authors must briefly present the adaptation of this method to the fungal cell wall and how this method was used for quantification of chitin oligomers (chit4, chit3, and chit2), N-acetyl-glucosamine, and glucose.
Response: indeed the reference is fracois 2006 and was added in the text and reference. Sorry about this mistake.
Reviewer 3 Report
As general comment the work is well written and designed with relevant results.
The authors touch upon very important issues about the Trichoderma reesei.
The issues discussed by the Authors are original.
This manuscript timely and I commend the Authors for bringing in some new ideas and analysis.
The work does not raise any scientific or substantive reservations.
This study is very interesting and conforms to the requirements of the Microorganisms journal.
Materials and method section is well described and correspond to the aim set out in the manuscript.
The statistical calculation methods used in the research make the obtained results reliable and provide a basis for drawing correct conclusions.
The results are correctly described.
The tables and figure are clear and understandable.
The discussion is correct.
The conclusions of the article are consistent with the problems discussed.
The references are properly chosen and cited.
The paper needs some editorial corrections.
I recommend the publication of this manuscript in the Microorganisms journal.
Author Response
Thanks for your comments.
Reviewer 4 Report
The work of Alinne Costa Silva et al., is sounded, the manuscript is well written and the output advanced our knowledge on understanding of the role of the EPL2 protein in T. reesei, enhancing its application in biotechnology fields. I only suggest to add more details on the sequencing methods.
Author Response
Thanks for your comments. We added more information and sequences about EPL4 as suggested by Reviewer 1
Reviewer 5 Report
The authors present an interesting paper uncovering genes affected by the Cerato-platanin Family protein (epl2) in Trichoderma reesei using transcriptomic analysis. It helps us understand how EPL2 involved in cellulose recognition processes and cell wall remodeling, enhancing its application in biotechnology fi elds. I recommend minor revision before acceptance.
In this paper, althought the authers found the up-regulations of CAZy genes in Δepl2 mutant, i think it will better for the authors to provide more data on extracellular enzyme activity to support their conclusions.
The general flow and the logic of the manuscript seems approriate. The English grammar and sentence structure are clear to read.
Author Response
Thanks for your comments.
We did not show the results about extracellular enzymes because we did not observe statistical differences between wild-type and mutant. This sentence was added to the text.
Round 2
Reviewer 1 Report
The work has some elegant studies and sheds new light on the CP protein's roles. However there are serious problems in the writing- many sentences are ambigous in meaning, some statements are in conflict with other statements, some words are incorrect
The information in supplemental lacks enough detail for some sections and the quality of image is poor

poor needs professional SCIENTIFIC editing - many sentences need to be examined and rewritten to make sure the sense is correct
just editing for words will not work
Reviewer 2 Report
The authors made the requested improvement. The manuscript is suitable for publication.
Author Response
Thanks for your consideration